# Association between Menopausal Hormone Therapy and Frailty: Cross-Sectional Study Using National Survey Data in Korea

**DOI:** 10.3390/healthcare10112121

**Published:** 2022-10-24

**Authors:** Hyunjoo Kim, Euni Lee

**Affiliations:** College of Pharmacy & Research Institute of Pharmaceutical Sciences, Seoul National University, 1 Gwanak-ro, Gwanak-gu, Seoul 08826, Korea

**Keywords:** cross-sectional studies, estrogen replacement therapy, frailty, hormone replacement therapy, women’s health

## Abstract

Frailty is a multidimensional clinical syndrome that increases the risk of adverse health outcomes. Previous studies have reported a close link between menopause and frailty. Combined estrogen–progestin therapy (or estrogen-only therapy in women who have undergone a hysterectomy) is currently approved as a menopausal hormone therapy (MHT) to treat menopausal symptoms. Despite increasing evidence of the importance of sex hormones in the development of frailty, very few studies have investigated the association between MHT and frailty. A cross-sectional evaluation was conducted using population-based survey data known as the Korea National Health and Nutrition Examination Survey (KNHANES IV-V, 2008–2012). The KNHANES data provided variables that were used to construct a 51-item frailty index (FI). The number of study population, only including postmenopausal women, was 7823 women, and their mean age was 62.51 years (range 32–80 years). Approximately 40% of them had graduated from middle school or higher, 45% lived in metropolitan statistical areas, and 5% were recipients of the national Medical Aid. The mean age at menopause was 48.66 years (range 30–62 years). Overall, the mean FI value was 0.15, and the prevalence of MHT was 13.23%. Findings from multiple regression analysis using the inverse probability of treatment weighting showed that a treatment duration of more than 2 years and up to 5 years, age at first treatment between 50 and 59 years, and MHT initiation 3 to 6 years after menopause were all negatively associated with frailty (*p* < 0.05). Further studies are needed to confirm these findings using prospective data.

## 1. Introduction

Frailty is a multidimensional and potentially preventable clinical syndrome characterized by the depletion of physiological reserves and resilience to stress [1,2,3]. Frailty is of concern to society, as frail individuals become prone to deleterious health outcomes including falls, hospitalization, and death [1,4,5]. Well-known risk factors for frailty include age, multimorbidity, poor socioeconomic status, and a higher number of pregnancies in women [6,7,8,9]. Although the prevalence of frailty increases in both men and women with age, it is predominantly higher in women, by up to twofold in the oldest old [10]. There is no definitive explanation for such sex differences; however, some studies have found the potential role of sex hormones as a key factor in women’s frailty [6,11]. 

Menopause, the permanent cessation of menstruation, is accompanied by a significant fall in blood estrogen [12]. Previous studies have reported a close link between age at menopause and frailty [8,9,13], which may partly be explained by the loss of the protective effects of estrogen on various physiological systems including lipid profiles, cognitive function, bone and muscle density, and inflammation [6,14,15]. In previous studies, MHT was associated with a lower prevalence of sarcopenia and increased grip strength, which are key criteria for physical frailty [16,17]. However, these associations remain unclear, as the findings from some studies have reported otherwise [9,18]. 

To treat menopausal symptoms, combined estrogen–progestin therapy (in women with an intact uterus) or estrogen-only therapy (in women who have undergone a hysterectomy) is currently approved as menopausal hormone therapy (MHT) [19]. Current MHT recommendations are largely governed by the “timing hypothesis”, which posits that MHT is effective only when administered near the start of menopause in younger postmenopausal women [20]. The hypothesis helps explain the discrepancies in the study outcomes between multiple observational studies and the Women’s Health Initiative (WHI) trial, as the WHI trial included many women over 60 years of age [21].

The protective effects of MHT against frailty remain controversial, and only a small number of studies have investigated the association, and even fewer studies have used national-level big data. Therefore, this study was conducted to investigate the association between MHT and frailty, specifically in terms of treatment duration, initiation age, and time to initiation after menopause. Additionally, we compared the demographic and socioeconomic characteristics between women using MHT and those who did not, and further investigated factors associated with frailty in postmenopausal Korean women. 

## 2. Materials and Methods

### 2.1. Participants

Women’s menopausal status was determined by the availability of the age at menopause. As the purpose of the study was to include all types of menopause, i.e., premature, early, normal, and late, participants with age at menopause in between 30 and 62 were included, similar to a previous study [9]. As data on hysterectomies were available only in 2008–2009 and data on bilateral oophorectomies were available only in 2010–2012, women with history of hysterectomy or bilateral oophorectomy were all included in the study. 

### 2.2. Instruments

We used a cross-sectional survey, Korea National Health and Nutritional Examination Survey (KNHANES), which is provided by the Korea Disease Control and Prevention Agency (KDCA). KNHANES assesses the health status of the non-institutionalized Korean population through health interviews, health examinations, and nutritional surveys [22]. The complex multistage clustered probability sampling design of the survey provides sampling weights allowing the generation of national-level estimates [22]. We used five consecutive years (2008–2012) of survey data (KNHANES IV, V). We used these years of data as the information on MHT duration and age at initiation, which were important attributes in our study, were not available after 2012. Furthermore, data from 2007 were excluded as they did not provide results from certain clinical examinations that were required in frailty assessments.

### 2.3. Procedures

KNHANES receives yearly approval from the KDCA Research Ethics Committee based on the relevant regulations, including the National Bioethics and Safety Act and the Declaration of Helsinki (IRB No. 2008-04EXP-01-C, 2009-01CON-03-2C, 2010-02CON-21-C, 2011-02CON-06-C, 2012-01EXP-01-2C) [22]. Consent from the participants was obtained by the KDCA’s appointed in-field investigators before the survey [22]. The protocol of this study, which used secondary data, was approved by the Institutional Review Board (IRB) of the Seoul National University Hospital (IRB No. E2111/002-003).

#### 2.3.1. Menopausal Hormone Therapy (Exposure Variable)

During the interviews, participants were asked, “*Have you ever taken oral hormone therapy for at least one month, excluding hormones taken for contraception?*”. Women who answered “*No*” were considered controls, and those who answered “*Yes*” were considered as treated. To avoid misclassification bias, women who initiated hormone therapy more than 5 years before their age at menopause were excluded [23]. Due to the inherent limitation of the database only capturing oral route of MHT, it was not possible to flag women who were prescribed with non-oral administration of MHT; therefore, they were included as controls.

In the treatment group, MHT-related information was collected using provided data or calculated as follows: the duration of MHT was provided as the sum of all recorded months when MHT was administered, age at first MHT administration was calculated by subtracting the difference between the year of the survey and the year of first MHT from the participant’s age at the survey (the information was calculated for 2008–2009 only, as it was provided in 2010–2012), and time to MHT initiation after menopause was calculated by subtracting the age at first MHT administration from the age at menopause. 

#### 2.3.2. Frailty Index (Outcome Variable)

The two most commonly used methods to operationalize frailty are the frailty index (FI), which is a ratio of health deficits such as symptoms, diseases, or laboratory values present in an individual; and the frailty phenotype, i.e., physical frailty, which determines an individual as frail when they have at least three of the following criteria: weight loss, weak grip strength, self-reported exhaustion, slow walking speed, and low physical activity [4,5,10]. Currently, there is no gold standard for measuring frailty; however, some studies have suggested that the FI has higher predictability for the risks of adverse outcomes [4,5].

A 51-item FI was constructed according to standard guidelines [5], as described in Appendix A, similar to previous frailty studies [24]. To capture the participants’ frailty as close as possible to the time of the survey, items reflecting the participants’ recent health status were used; for example, a history of hospitalization was not used in the FI calculation, as the history within the entire year was collected. 

### 2.4. Data Analysis

Descriptive statistics were presented to summarize the participants’ overall characteristics and compare the control and treatment groups. Regression analyses were conducted to investigate the associations between the MHT-related information and frailty, with the control group as the reference variable. 

To control for confounding, the propensity score (PS) based inverse probability of treatment weighting (IPTW) was applied to estimate the average treatment effect (ATE) at a national level [25,26]. Stabilized IPTW weights were generated using prognostically or clinically important information such as socioeconomic, reproductive, and clinical characteristics [7,8,27]. Variables to be included in the PS model were selected from those significantly different between the treatment groups. Additionally, clinical conditions listed as contraindications or warned conditions in the approved product labels of MHT were included in the PS model as they could affect treatment decisions. The conditions were coded to reflect the participant’s status at baseline, i.e., before initiating MHT (treated) or before menopause (controls), using the “age at diagnosis” information. 

After IPTW, the standardized mean differences (SMD) between the treatment group and the controls, and the variance ratios were well within the recommended ranges (Appendix A) [28]. To test the robustness of the results, a sensitivity analysis was conducted by limiting the study population to women who had been in menopause for at least 1 year and less than 30 years [29]. Moreover, to account for heterogeneous types of menopause, subgroup analyses were conducted in women aged over 45 years at menopause [9], women with an intact uterus (data available only in the years 2008–2009), and women with no history of bilateral oophorectomy (data available only in the years 2010–2012). 

All statistical analyses were performed using SAS version 9.4 (SAS Institute Inc., Cary, NC, USA), and the level of statistical significance was set at *p* < 0.05 (two-sided).

## 3. Results

In total, 23,037 women participated in KNHANES from 2008 to 2012; of these, 8010 were postmenopausal and had data available on their history of MHT. In the final study population, 7823 postmenopausal women in total for whom the weights were available for IPTW were included (controls: 6779; treated: 1044) (Figure 1). 

The mean FI was 0.15, which is a number within the pre-frail range (0.08 < FI < 0.25) [30]. The mean age of the study population was 62.51 years, with a range from 32 to 80 years. Overall, 65.74% participated in the survey’s Cycle V (2010–2012). Approximately 40% had graduated from middle school or higher, 33% received the lowest quartile of household income, 45% lived in metropolitan statistical areas (MSAs), and 5% were recipients of the national Medical Aid. Many (67%) women were married and living together. Overall, the mean age at menopause was 48.66 years, with a range from 30 to 62 years, and more than 10% had started menopause at least 30 years ago (Table 1).

In total, 13.23% were treated with MHT. At the time of the survey, compared with women in the control group, women in the treatment group were significantly less frail (FI: 0.13 vs. 0.15); were younger (58.45 vs. 63.13 years old); were more likely to be in the survey’s Cycle V (2010–2012); had a higher socioeconomic status in terms of education, household income, living area, and type of national insurance; and more were married and living together (all *p* < 0.05) (Table 2). 

In terms of reproductive history, women who were treated had a significantly lower number of pregnancies and birth experiences, a longer duration of oral contraceptives (OC) use, and a slightly younger age at menarche (all *p* < 0.01). Additionally, the number of comorbidities at baseline, i.e., before initiating MHT (treated) or before menopause (controls), was significantly higher in the treatment group (*p* < 0.001). By conditions, the baseline prevalence of hypertension, dyslipidemia, depression, thyroid illness (both hyper- and hypothyroidism), and arthritis (rheumatoid arthritis and osteoarthritis) were significantly higher in the treatment group (all *p* < 0.05) (Table 2), even after stratification by age at baseline (*p* < 0.001; Appendix A). 

After IPTW, most differences were no longer statistically significant (Appendix A). Those with remaining statistical significance were additionally adjusted in the subsequent regression analyses.

In women who were treated, the median duration of MHT was 15.23 months, ranging from 1 month to 36 years. The mean age at first treatment was 50.79 years, and many (69.31%) initiated treatment within 2 years after menopause (Table 3). 

From the IPTW univariable regression analysis, older age, 2008–2009 survey year, lower socioeconomic status, living alone, smoking history, older age at menarche, younger age at menopause, a higher number of pregnancies, and baseline multimorbidity were all positively associated with frailty (*p* < 0.01 for all) (Table 4). 

From the IPTW multiple regression analysis, a treatment duration of more than 2 years to 5 years or less, age at first treatment between 50 and less than 60 years, and MHT initiation 3 to 6 years after menopause were all negatively associated with frailty (*p* < 0.05 for all; Model 3) (Table 5). The regression estimates for other variables are provided in Appendix A.

Subgroup analyses in KNHANES yielded similar results to those in the overall population (Table 6). The characteristics of the subgroups are provided in Appendix A.

## 4. Discussion

In this study, we found significant associations between frailty and MHT in terms of treatment duration, initiation age, and time to initiation after menopause in postmenopausal Korean women. The mean FI of the study population was 0.15, an estimate that is well within the range of those previously reported in middle-aged women [11,13], and approximately 13% had a history of receiving MHT. When FI was categorized using a previously reported cutoff value of 0.25 or higher [30], the prevalence of frailty in women who were treated was approximately half of that in women who were never treated (6.23% vs. 12.72%; data not provided). To the best of our knowledge, this was the first study to report these associations using nationwide population-based survey data. These estimates provide valuable insights into managing frailty in postmenopausal Asian women.

There is increasing evidence of the associations between MHT and physical frailty. In a previous cross-sectional study in Korea (KNHANES 2008–2011), compared with those never treated or those who had received treatment for <13 months, a longer duration of MHT (≥13 months) was significantly associated with lower odds of sarcopenia, a criterion for physical frailty [16]. In a cross-sectional analysis of an ongoing cohort study (the Korean Frailty and Aging Cohort Study), where women aged 70 to 84 years were enrolled, the prevalence of physical frailty was lower in women with a history of MHT, approximately one-third of what was observed in those with no history of MHT [8]. However, there is also evidence that contradicts these findings. For example, in the WHI trial, MHT did not protect against physical frailty after 6 years of follow-up in women who were at least 65 years old [31]. In a cross-sectional analysis of a cohort study (the Canadian Longitudinal Study on Aging, 2012–2015), where adults between the ages of 45 and 85 were enrolled, the mean FI was significantly higher in women with a history of MHT (0.12 vs. 0.11) [9]. The differences in the reported associations may be due to the heterogeneity in the women’s age, the study design, or racial disparities in MHT’s effectiveness, which should be further investigated in future studies.

Certain characteristics were significantly related to MHT use. Women of higher socioeconomic status were more likely to use MHT; whereas women living in rural areas, living alone, and who had had a higher number of pregnancies or birth experiences were less likely to use MHT. Such findings were all consistent with previous findings [32,33,34]. Previously, postmenopausal Korean women were interviewed about their reasons for not using MHT. More than half of the participants (55.0%) answered they did not know about MHT; other reasons were having no menopausal symptoms (15.6%), the expensive cost (8.9%), concerns about the side effects (8.7%), and limitations of distance and time for accessing medical services (6.9%) [34]. Therefore, the characteristics mentioned above may have affected the patients’ access to healthcare or health-related information, and their economic or time resources, which could have ultimately influenced their treatment decisions. Additionally, the mean duration of previous OC use was longer in the treatment group, suggesting that women with more experience in using hormone therapy are more likely to use MHT [33].

Based on common practice, women with contraindications or warned conditions against the use of MHT are unlikely to be treated with MHT. One of the concerns about this study was that many women in the control group would have such conditions and therefore would have a higher FI value. In such a case, biased results would be obtained. Therefore, we decided to compare the baseline prevalence of these conditions between the groups. The results revealed that the prevalence of contraindications or warned conditions was higher in the treatment group. Possible reasons for this finding may be the earlier detection of chronic diseases as a result of active surveillance by scheduled visits to healthcare services, or a greater need for hormone therapy for health-related reasons in the treatment group. It may also be that the physicians lack access to the patients’ comprehensive medical history, or despite such conditions, the physicians still may have decided to prescribe MHT due to preferable benefit–risk profile, accompanied by close monitoring. Of note, as KNHANES was not a clinical database, we were unable to screen all the contraindications or warned conditions.

The results from our multiple regression analyses are in line with current MHT recommendations governed by the “timing hypothesis” [20], which supports a favorable benefit–risk ratio when MHT is provided near the start of menopause in younger menopausal women, preferably before the age of 60 or within 10 years of menopause [27,35]. Our results consistently demonstrated such a trend, where frailty was significantly lower in women who initiated MHT before the age of 60. A negative association was also observed when MHT was initiated between 3 and 6 years after menopause, which is approximately the period when postmenopausal estradiol values begin to stabilize [36]. An association was also found with an MHT duration of 2 to 5 years, which is in alignment with current evidence, where an increased risk of venous thromboembolism was reported in the first 1–2 years of treatment, and an increased risk of breast and ovarian cancers with treatment periods longer than 5 years [35,37,38]. Additionally, older age, low socioeconomic status, baseline multimorbidity, younger age at menopause, and a higher number of pregnancies were all positively associated with frailty, which are all consistent with previous findings [6,7,8,9,13]. In the subgroup analysis, similar results were found as in the overall analysis. Of note, some inconsistencies were observed in the “no hysterectomy” group, which may stem from an insufficient sample size, i.e., at least 30 per cell for complex survey data [39]. In terms of the time to initiation after menopause, the association between frailty and treatment initiation 3–6 years after menopause was lost.

Estrogen may be a key factor in women’s frailty [6,11]. However, the evidence suggested by the majority of current research has focused on the effect of estrogen on frailty in terms of musculoskeletal function, such as sarcopenia and grip strength [13]. We believe that the findings from our study could provide an expanded understanding of the role of MHT in frailty from the perspective of biological plausibility, as the construction of the frailty index in our study accounted for various laboratory values, which may allow the detection of accumulated subclinical deficits [40]. A growing body of evidence has indicated the crucial role of estrogen in the regulation of inflammation [14]. For example, decreased ovarian function in postmenopausal women was associated with increased levels of inflammatory cytokines such as IL-6, TNF-a, and C-reactive protein [6]. In previous in vitro and in vivo studies, reduced estrogen was associated with excessive inflammation and delayed healing, specifically via the increased release of TNFα and macrophage migration inhibitory factor (MIF) [14]. As inflammation plays a crucial role in the development of frailty [13], MHT may exert protective effects against frailty via the inhibition of subcellular pro-inflammatory pathways. As frailty involves complex biological processes and the associated changes are subtle, further studies are needed to clarify the underlying mechanisms linking estrogen and frailty.

Our study has many of the limitations inherent to cross-sectional secondary survey data, including narrow room for causal inference. Moreover, as the most recent data from KNHANES were from 2012, data from more recent years may have provided different results. Second, when KNHANES data was collected, women who took MHT for less than a month were coded as “never treated”. As it was not possible to discern absolute, non-exposed patients, we investigated the association between MHT and frailty with varying duration of treatment. Third, it would have been clinically meaningful if MHT administered via non-oral routes, e.g., transdermal, had been included in the analysis. We can only infer these from a previous study in Korea, where approximately 13% among treated were administering MHT via non-oral routes between 2002 and 2013 [32]. Based on the study, we projected that approximately 1.7% of the study population would be potentially misclassified as the control group. We believe the magnitude of the potential misclassification bias impacting our results is minimal as the direction of the bias would be toward the null. Fourth, we were unable to flag women with a history of hysterectomy or bilateral oophorectomy due to insufficient data, which would have affected the MHT combination, i.e., the presence or absence of progesterone. As the information on MHT combination was unavailable in KNHANES, we conducted additional sensitivity analyses by excluding women with hysterectomy in KNHANES IV and bilateral oophorectomy in KNHANES V. We believe the magnitude of the potential bias due to hysterectomy or bilateral oophorectomy would be minimal as the national use of estrogen monotherapy was 7% in 2010, which projects to less than 1% of the study population [41]. Future studies should further investigate the association between frailty and MHT with varying dosage, active substances and combinations, and routes of administration. Lastly, our study should be interpreted with caution as most of the study data, including MHT-related information and the age at diagnosis of each medical condition, were reported by participants and therefore prone to recall bias. To mitigate the potential recall bias, we conducted a sensitivity analysis by excluding women with menopause for more than 30 years.

The main strengths of our study were as follows. First, the use of IPTW allowed a balance between the control and treatment groups. Second, to minimize inverse causality, the variables used in the frailty assessments were coded to reflect recent health status near the time of survey participation, whereas medical histories in the PS model were coded to reflect the baseline health status. Third, a series of subgroup analyses were conducted to account for possible recall bias and to consider heterogeneous types of menopause, which all yielded similar results to the main analysis in KNHANES. Lastly, we assessed MHT in terms of duration, age at initiation, and time to initiation after menopause, allowing the study to be interpreted within the context of the current treatment guidelines.

In summary, our study successfully demonstrated a potential beneficial effect of MHT, when provided within an appropriate treatment window, for managing frailty in middle-aged Asian women. These results should be further validated using prospective cohort studies.

## 5. Conclusions

Our study findings highlighted the negative associations between frailty and MHT in terms of treatment duration of more than 2 and up to 5 years, initiation before the age of 60, and treatment initiation within 3 to 6 years after menopause. MHT may be beneficial for promoting healthy aging when provided within an appropriate treatment window in postmenopausal Asian women. Further investigation is required using prospective data.

## Figures and Tables

**Figure 1 healthcare-10-02121-f001:**
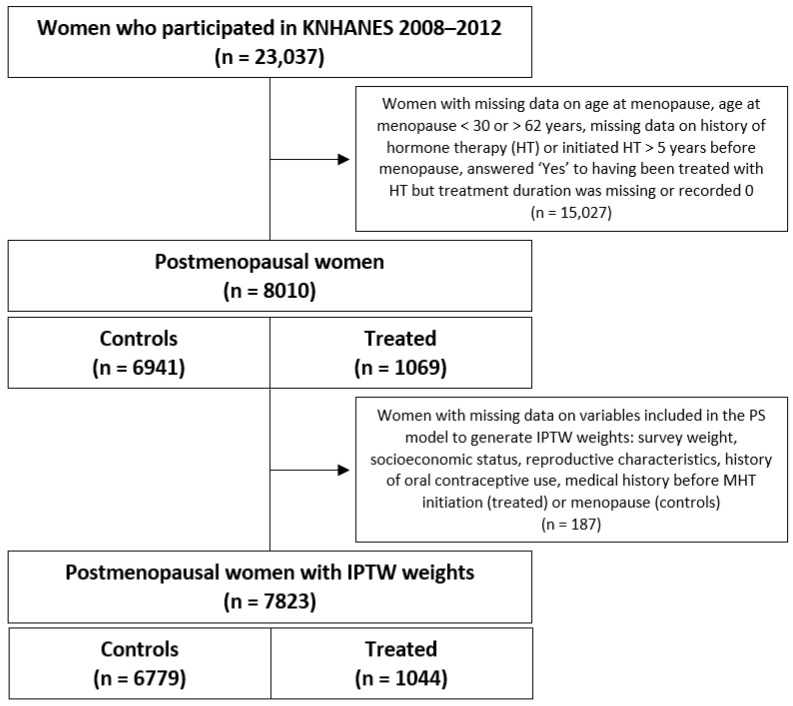
Identification of study population within KNHANES participants (2008–2012).

**Table 1 healthcare-10-02121-t001:** Characteristics of study population.

Characteristics	Total		
	*n*	*n’*	% (SE)
Total	7823	4289	13.48 (0.23)
Frailty index *	0.15 (0.00)		
Age at survey *	62.51 (0.15)		
Survey year: 2008–2009	2846	1762	34.26 (0.80)
2010–2012	4977	2810	65.74 (0.80)
Education: <Middle school	4949	2462	59.72 (0.85)
≥Middle school	2874	1919	40.28 (0.85)
Household income: Not low	4955	3359	67.33 (0.76)
Low ^a^	2868	1612	32.67 (0.76)
Living area: MSA ^b^	3251	1641	44.78 (1.06)
Other	4572	1876	55.22 (1.06)
Married and living together: Yes	5145	3169	66.74 (0.72)
No ^c^	2670	1809	33.26 (0.72)
Insurance: Self/company	7361	4119	94.68 (0.33)
Other ^d^	462	371	5.32 (0.33)
Smoking history: No	7181	4520	90.77 (0.44)
Yes	636	511	9.23 (0.44)
Number of pregnancies *	4.77 (0.035)		
Birth experience: 0	165	135	2.09 (0.20)
1 or more	7626	3727	97.91 (0.20)
Months of OC use *	4.55 (0.23)		
Age at menarche *	15.95 (0.03)		
Age at menopause *	48.66 (0.07)		
Time since menopause: <10 years	2845	1475	43.62 (0.78)
10–19 years	2253	1582	26.22 (0.60)
20–29 years	1787	1335	19.38 (0.52)
≥30 years	914	644	10.78 (0.42)

Inverse probability of treatment weighting was not applied. * mean (SE). *n*, unweighted frequency; *n’*, effective sample size (=*n*/design effect); %, weighted percentage; SE, standard error; MSA, metropolitan statistical areas; OC, oral contraceptives. ^a^ Equivalized household income, i.e., household income/√ (# of household members) in the lowest 25%; quartiles were stratified by sex and age group. ^b^ Metropolitan statistical areas, i.e., top eight major cities in the Republic of Korea. ^c^ Never married, separated, widowed, or divorced. ^d^ Medical Aid Class 1 or 2, no health insurance, or unknown.

**Table 2 healthcare-10-02121-t002:** Comparison of characteristics between treatment groups.

Characteristics	Control			Treated			*p*
	*n*	*n’*	% (SE)	*n*	*n’*	% (SE)	
Total	6779	3393	86.77 (0.49)	1044	795	13.23 (0.49)	-
Frailty index *	0.15 (0.001)			0.13 (0.003)			**<0.001**
Age at survey *	63.13 (0.17)			58.45 (0.25)			**<0.001**
Survey year: 2008–2009	2555	1302	34.90 (0.83)	291	166	30.05 (1.89)	**0.016**
2010–2012	4224	1981	65.10 (0.83)	753	464	69.95 (1.89)	
Education: <Middle school	4503	2056	62.46 (0.90)	446	278	41.74 (1.93)	**<0.001**
≥Middle school	2276	997	37.54 (0.90)	598	374	58.26 (1.93)	
Household income: Not low	4129	2159	65.31 (0.83)	826	529	80.58 (1.45)	**<0.001**
Low ^a^	2650	1294	34.69 (0.83)	218	153	19.42 (1.45)	
Living area: MSA ^b^	2705	836	43.56 (1.12)	546	355	52.80 (2.07)	**<0.001**
Other	4074	1267	56.44 (1.12)	498	266	47.20 (2.07)	
Married and living together: Yes	4322	2416	64.99 (0.77)	823	514	78.22 (1.59)	**<0.001**
No ^c^	2450	1405	35.01 (0.77)	220	143	21.78 (1.59)	
Insurance: Self/company	6345	4027	94.29 (0.37)	1016	633	97.26 (0.60)	**0.001**
Others ^d^	434	256	5.71 (0.37)	28	20	2.74 (0.60)	
Smoking history: No	6206	3668	90.57 (0.48)	975	617	92.08 (1.06)	**0.220**
Yes	567	309	9.43 (0.48)	69	42	7.92 (1.06)	
Number of pregnancies *	4.82 (0.04)			4.42 (0.08)			**<0.001**
Birth experience: 0	141	101	2.08 (0.21)	24	16	2.13 (0.55)	**0.009**
1 or more	6610	3882	97.92 (0.21)	1016	632	97.87 (0.55)	
Months of OC use *	4.22 (0.25)			6.75 (0.65)			**<0.001**
Age at menarche *	15.99 (0.04)			15.70 (0.07)			**<0.001**
Age at menopause *	48.69 (0.08)			48.43 (0.17)			0.153
Baseline comorbidities ^e^: 0	4078	2826	59.38 (0.73)	491	340	46.29 (1.87)	**<0.001**
1	1958	1400	29.13 (0.65)	364	258	34.68 (1.72)	
≥2	743	443	11.49 (0.49)	189	121	19.02 (1.46)	
Hypertension: No	6281	3856	92.09 (0.42)	910	596	86.68 (1.26)	**<0.001**
Yes	498	298	7.91 (0.42)	134	89	13.32 (1.26)	
Dyslipidemia: No	6598	4034	97.12 (0.26)	992	625	95.59 (0.72)	**0.023**
Yes	181	112	2.88 (0.26)	52	41	4.41 (0.72)	
MI: No	9432	7541	99.97 (0.02)	479	381	100.0 (0.00)	0.806
Yes	3	3	0.03 (0.02)	0	0	.	
AP: No	6718	4261	99.00 (0.14)	1031	647	98.75 (0.43)	0.562
Yes	61	43	1.00 (0.14)	13	8	1.25 (0.43)	
Stroke: No	6730	4241	99.35 (0.11)	1033	638	99.16 (0.31)	0.534
Yes	49	38	0.65 (0.11)	11	9	0.84 (0.31)	
Diabetes: No	6598	3980	97.07 (0.28)	1009	627	96.81 (0.64)	0.704
Yes	181	100	2.93 (0.28)	35	26	3.19 (0.64)	
Hepatitis B/C, cirrhosis: No	6704	4225	98.86 (0.16)	1029	637	98.43 (0.46)	0.329
Yes	75	52	1.14 (0.16)	15	11	1.57 (0.46)	
Cancer: No	6682	4314	98.37 (0.20)	1013	622	97.51 (0.53)	0.081
Yes	97	58	1.63 (0.20)	31	26	2.49 (0.53)	
Renal failure: No	6759	4203	99.72 (0.08)	1040	647	99.67 (0.17)	0.774
Yes	20	15	0.28 (0.08)	4	5	0.33 (0.17)	
Depression: No	5542	3642	82.25 (0.57)	779	508	72.95 (1.75)	**<0.001**
Yes	1237	798	17.75 (0.57)	265	158	27.05 (1.75)	
Thyroid illness: No	6598	4365	96.92 (0.28)	995	636	95.01 (0. 82)	**0.011**
Yes	181	99	3.08 (0.28)	49	33	4.99 (0. 82)	
Asthma: No	6584	4113	97.01 (0.28)	1008	645	96.05 (0.76)	0.203
Yes	195	104	2.99 (0.28)	36	22	3.95 (0.76)	
Arthritis: No	5869	4005	86.26 (0.53)	868	575	83.00 (1.40)	**0.017**
Yes	910	557	13.74 (0.53)	176	115	17.00 (1.40)	

Inverse probability of treatment weighting was not applied. * mean (SE). *n*, unweighted frequency; *n’*, effective sample size (=*n*/design effect); %, weighted percentage; SE, standard error; MSA, metropolitan statistical areas; OC, oral contraceptives. ^a^ Equivalized household income, i.e., household income/√(# of household members) in the lowest 25%; quartiles were stratified by sex and age group. ^b^ Metropolitan statistical areas, i.e., top eight major cities in the Republic of Korea. ^c^ Never married, separated, widowed, or divorced. ^d^ Medical Aid Class 1 or 2, no health insurance, or unknown. ^e^ Ever diagnosed before menopause (controls) or MHT (treatment group): hypertension, dyslipidemia, myocardial infarction (MI), angina pectoris (AP), stroke, diabetes, liver disease (hepatitis B or C, cirrhosis), cancer excluding skin cancer, renal failure, depression, thyroid illness (both hyper- and hypothyroidism), asthma, or arthritis (rheumatoid arthritis and osteoarthritis).

**Table 3 healthcare-10-02121-t003:** MHT-related information in treatment group.

Characteristics	Treated		
	*n*	*n’*	% (SE)
Duration of MHT (months)			
Median (IQR)	15.23 (3.20–39.70)		
Min–Max	1–432		
<6 months	300	207	28.88 (1.65)
6 months–2 years	345	215	34.49 (1.81)
>2, ≤5 years	252	160	24.13 (1.66)
>5, ≤8 years	72	54	6.69 (0.91)
>8 years	75	69	5.82 (0.76)
Age at first MHT (years)			
Mean (SE)	50.79 (0.20)		
Min–Max	31–72		
<50	356	279	34.95 (2.13)
≥50, <55	443	355	43.90 (2.25)
≥55, <60	171	165	14.22 (1.47)
≥60	74	82	6.93 (1.08)
Time to initiate MHT (years)			
Median (IQR)	0.31 (−0.60 to 2.91)		
Min–Max	−5 to 28		
Before menopause	113	82	11.46 (1.16)
<1 year after menopause	342	213	34.04 (1.79)
1–2 years after menopause	237	156	23.81 (1.62)
3–6 years after menopause	191	131	17.79 (1.42)
7–10 years after menopause	90	77	7.18 (0.88)
≥11 years after menopause	71	67	5.72 (0.74)

Inverse probability of treatment weighting was not applied. *n*, unweighted frequency; *n’*, effective sample size (=*n*/design effect); %, weighted percentage; SE, standard error of weighted %; IQR, interquartile range; MHT, menopausal hormone therapy.

**Table 4 healthcare-10-02121-t004:** Characteristics associated with frailty (IPTW regression analysis).

Characteristics (Reference)	B (SE)	*p*
Age at survey	0.004 (0.000)	**<0.001**
Survey year, 2010–2012 (2008–2009)	−0.010 (0.003)	**0.004**
Middle school, graduated (did not graduate)	−0.045 (0.003)	**<0.001**
Household income, low ^a^ (not low)	0.043 (0.004)	**<0.001**
Living area, other (MSA ^b^)	0.015 (0.003)	**<0.001**
Married and living together, no ^c^ (yes)	0.037 (0.003)	**<0.001**
National insurance, others ^d^ (self/company)	0.056 (0.007)	**<0.001**
Smoking history, yes (no)	0.019 (0.006)	**0.001**
Age at menarche	0.003 (0.001)	**<0.001**
Age at menopause	−0.001 (0.000)	**0.004**
Number of pregnancies	0.007 (0.001)	**<0.001**
Birth experience, 1 or more (0)	0.009 (0.007)	0.235
Months of oral contraceptives use	0.000 (0.000)	0.059
Number of comorbidities at baseline ^e^: 1 (0)	0.017 (0.003)	**<0.001**
≥2 (0)	0.050 (0.005)	**<0.001**

Inverse probability of treatment weighting (IPTW) was applied. SE, standard error; MHT, menopausal hormone therapy. ^a^ Equivalized household income, i.e., household income/√(number of household members) in the lowest 25%; quartiles were stratified by sex and age group. ^b^ Metropolitan statistical areas, i.e., top eight major cities in the Republic of Korea. ^c^ Never married, separated, widowed, or divorced. ^d^ Medical Aid Class 1 or 2, no health insurance, or unknown. ^e^ Ever diagnosed before menopause (controls) or MHT (treatment group): hypertension, dyslipidemia, myocardial infarction (MI), angina pectoris (AP), stroke, diabetes, liver disease (hepatitis B or C, cirrhosis), cancer excluding skin cancer, renal failure, depression, thyroid illness (both hyper- and hypothyroidism), asthma, and arthritis (rheumatoid arthritis and osteoarthritis).

**Table 5 healthcare-10-02121-t005:** Associations between MHT-related information and frailty (IPTW regression analysis).

Characteristics (Reference)	Model 1 ^a^	Model 2 ^b^	Model 3 ^c^
	B (SE)	B (SE)	B (SE)
Duration of MHT			
<6 months (control)	−0.009 (0.006)	0.001 (0.005)	−0.003 (0.005)
6 months–2 years (control)	**−0.019 (0.005)**	−0.006 (0.005)	−0.008 (0.004)
>2, ≤5 years (control)	**−0.020 (0.006)**	**−0.013 (0.005)**	**−0.015 (0.005)**
>5, ≤8 years (control)	−0.005 (0.014)	0.021 (0.017)	0.018 (0.016)
>8 years (control)	−0.012 (0.014)	−0.009 (0.012)	−0.012 (0.013)
Age at first MHT administration			
<50 (control)	**−0.015 (0.005)**	**0.013 (0.006)**	0.006 (0.005)
≥50, <55 (control)	**−0.022 (0.005)**	**−0.010 (0.004)**	**−0.012 (0.004)**
≥55, <60 (control)	−0.014 (0.008)	**−0.023 (0.007)**	**−0.020 (0.007)**
≥60 (control)	**0.028 (0.010)**	−0.007 (0.008)	−0.006 (0.009)
Time to MHT initiation after menopause			
Before menopause (control)	**−0.034 (0.007)**	−0.010 (0.008)	−0.009 (0.008)
<1 year (control)	**−0.017 (0.005)**	0.003 (0.006)	0.000 (0.006)
1–2 years (control)	−0.014 (0.007)	−0.003 (0.006)	−0.005 (0.005)
3–6 years (control)	**−0.018 (0.007)**	−0.012 (0.006)	**−0.014 (0.006)**
7–10 years (control)	0.001 (0.008)	−0.004 (0.007)	−0.009 (0.007)
≥11 years (control)	0.011 (0.010)	−0.013 (0.008)	**−0.021 (0.009)**

Inverse probability of treatment weighting (IPTW) was applied. SE, standard error; MHT, menopausal hormone therapy. Note: Bold text indicates the statistical significance of the estimates. ^a^ Model 1: Frailty index = MHT. ^b^ Model 2: Model 1 additionally adjusted for variables with statistical difference remaining after IPTW between the two groups. Frailty index = MHT + age at survey, survey cycle (IV or V), living area (metropolitan statistical area or others), marital status (married and living together vs. other), number of baseline comorbidities (0, 1, or ≥2), and history of depression at baseline. ^c^ Model 3: Model 2 additionally adjusted for variables associated with frailty, education (graduated from or did not graduate middle school), household income (low or not low), type of national insurance (company/self or other), smoking history (no or yes), age at menopause, age at menarche, and number of pregnancies.

**Table 6 healthcare-10-02121-t006:** Associations between MHT-related information and frailty (IPTW regression analysis, subgroups).

Characteristics	Years Since Menopause [1, 30)	Age at Menopause > 45	No Hysterectomy ^a^	No Bilateral Oophorectomy ^b^
	B (SE)	B (SE)	B (SE)	B (SE)
Duration of MHT (reference = control)				
<6 months	−0.001 (0.005)	0.001 (0.005)	0.003 (0.007)	−0.001 (0.006)
6 months–2 years	−0.006 (0.005)	−0.005 (0.005)	−0.013 (0.008)	−0.006 (0.006)
>2, ≤5 years	**−0.015 (0.005)**	**−0.016 (0.005)**	−0.007 (0.009)	**−0.016 (0.007)**
>5, ≤8 years	0.018 (0.016)	−0.005 (0.011)	0.007 (0.017)	0.007 (0.010)
>8 years	−0.010 (0.013)	−0.016 (0.016)	−0.051 (0.028)	−0.005 (0.014)
Age at first MHT administration (reference = control)				
<50	0.009 (0.005)	0.010 (0.006)	−0.006 (0.008)	**0.013 (0.006)**
≥50, <55	**−0.011 (0.004)**	**−0.010 (0.004)**	**−0.016 (0.008)**	**−0.011 (0.005)**
≥55, <60	**−0.019 (0.007)**	**−0.020 (0.007)**	−0.002 (0.013)	**−0.023 (0.008)**
≥60	−0.003 (0.009)	−0.003 (0.009)	0.011 (0.011)	−0.014 (0.003)
Time to MHT initiation after menopause (reference = control)				
Before menopause	−0.008 (0.009)	−0.006 (0.008)	**−0.025 (0.012)**	0.006 (0.009)
At menopause	0.003 (0.006)	0.001 (0.005)	−0.015 (0.008)	0.004 (0.006)
After 1–2 years	−0.004 (0.005)	−0.004 (0.006)	0.003 (0.010)	−0.007 (0.007)
After 3–6 years	**−0.014 (0.006)**	−0.023 (0.006)	−0.012 (0.011)	**−0.016 (0.007)**
After 7–10 years	−0.009 (0.008)	−0.004 (0.010)	0.015 (0.013)	−0.018 (0.009)
After ≥11 years	−0.019 (0.010)	−0.016 (0.011)	0.002 (0.012)	−0.026 (0.013)

Inverse probability of treatment weighting (IPTW) was applied. SE, standard error; MHT, menopausal hormone therapy. Note: Bold text indicates the statistical significance of the estimates. Model: Frailty index = MHT + age at survey, survey cycle (IV or V), living area (metropolitan statistical area or other), marital status (married and living together vs. other), number of baseline comorbidities (0, 1, or ≥2), history of depression at baseline, education (graduated from or did not graduate middle school), household income (low or not low), type of national insurance (company/self or other), smoking history (no or yes), age at menopause, age at menarche, and number of pregnancies. ^a^ Data on hysterectomies were collected in 2008 and 2009 only. ^b^ Data on bilateral oophorectomies were collected in 2010, 2011, and 2012 only.

## Data Availability

Data used in this study are publicly available for research purposes in a de-identified fashion at the official website of KNHANES: https://knhanes.kdca.go.kr/knhanes/eng/index.do (accessed on 22 October 2022).

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
