# Peer review of "Association between Menopausal Hormone Therapy and Frailty: Cross-Sectional Study Using National Survey Data in Korea"

_healthcare, 2022, doi:10.3390/healthcare10112121_

Round 1

Reviewer 2 Report

Dear Authors,

The manuscript presented for review raises important issues. However, some points need to be clarified:

- such a large study has only 2 authors?

- why were women who used the therapy by a different route (e.g. transdermal) included in the control group?

- why were women who used the therapy for even 1 month classified?

- why such a low age limit, ie 30 years, was adopted?

- the data comes from 2012 - why do the authors want to publish them only now? A lot of changes have happened in 10 years,

- there were 23,037 participants, 15,006 were rejected, so there are 8010 left? Or 8031?

- why is there such a disproportion in the size of the groups: controls (n = 6779) and treated (n = 1044)? Does this not affect the statistical conclusions?

- table 1 should be reorganized - there are different types of data that must be presented in a legible way,

- where are the IQR values ​​in the table? Should IQR not be reported at medians?

- how do the authors explain the point in line 131 - that living a partner has a significant impact?

- the discussion should explain the obtained results more clearly.,

- supplementary materials should be prepared in accordance with the MDPI guidelines.

Additionally:

- formatting must be in accordance with MDPI guidelines, e.g. information about the authors (initials, telephone number).

Reviewer 3 Report

The manuscript entitled "Association between Menopausal Hormone Therapy and Frailty: A Cross-sectional Study Using National Survey Data in Korea" addresses a topic of interest that may have a notable impact on the field, as it has, among other factors, a large database on which to perform complex analyses with great generalizability. Having said this, the following are a series of aspects that may be useful to reflect on or improve the manuscript, if necessary:

TITLE

Adequate. It provides information that is timely and at the same time defines the sample from which the information is going to be extracted so that the reader knows the origin from the beginning.

ABSTRACT

Provides relevant information. It would be interesting to include more sociodemographic data, specifically, the percentage of women or men and the mean age or range. 

KEYWORDS

It should be in alphabetical order and one of them should refer to the design followed.

INTRODUCTION

Again, it provides relevant information, but it is scarce. Given the size of the study, it is necessary to broaden the theoretical framework and go into more detail. It is also necessary to include a greater number of bibliographical references.

At the end of the introduction, the general and specific objectives should be made clearer. 

METHOD

The use of a large database is a very good resource to be able to work with the information and return to society through the publication of the results useful knowledge to know the reality of the subject and to be able to make decisions accordingly. Having said this, several questions arise that may help to clarify the procedure followed: How was consent requested from the Entity or Organization? When it is said that it was approved, does this mean that the ethical aspects of the study were also respected? Did it go through an Ethics Committee or did the Evaluation Committee itself have competence in ethical matters? Were the Helsinki Protocol Guidelines for human experimentation followed?

When describing the sample, a summary table with the main sociodemographic data should be included in this subsection. 

The recommended structure for the methodology section is: Participants, Instruments, Procedures and Data Analysis.

The "Statistical Analysis" section should be included in the methodology itself, not as a complete section.

RESULTS

Provides relevant information. However, it is necessary to previously define an internal structure of the results based on the specific objectives set out in the introduction (which should be included as mentioned above). 

DISCUSSION

An in-depth discussion of the subject matter is carried out, although there are some aspects that could be improved:

- Further develop future lines of research.

- Generate a different section called Conclusions and expand the final paragraph. 

- Make greater use of the references used in the Introduction. 

That said, this is a very interesting manuscript in which one can see that a considerable amount of time has been invested to include so much detail. 

Round 2

Reviewer 2 Report

The authors responded to all my questions contained in the review.